# MetaInit: Initializing learning by learning to initialize

**Yann N. Dauphin**
Google AI
ynd@google.com

**Samuel S. Schoenholz**
Google AI
schsam@google.com

## Abstract

Deep learning models frequently trade handcrafted features for deep features learned with much less human intervention using gradient descent. While this paradigm has been enormously successful, deep networks are often difficult to train and performance can depend crucially on the initial choice of parameters. In this work, we introduce an algorithm called MetaInit as a step towards automating the search for good initializations using meta-learning. Our approach is based on a hypothesis that good initializations make gradient descent easier by starting in regions that look locally linear with minimal second order effects. We formalize this notion via a quantity that we call the *gradient quotient*, which can be computed with any architecture or dataset. MetaInit minimizes this quantity efficiently by using gradient descent to tune the norms of the initial weight matrices. We conduct experiments on plain and residual networks and show that the algorithm can automatically recover from a class of bad initializations. MetaInit allows us to train networks and achieve performance competitive with the state-of-the-art without batch normalization or residual connections. In particular, we find that this approach outperforms normalization for networks without skip connections on CIFAR-10 and can scale to Resnet-50 models on Imagenet.

## 1 Introduction

Deep learning has led to significant advances across a wide range of domains including translation [55], computer vision [24], and medicine [2]. This progress has frequently come alongside architectural innovations such as convolutions [33], skip-connections [26, 22] and normalization methods [27, 4]. These components allow for the replacement of shallow models with hand-engineered features by deeper, larger, and more expressive neural networks that learn to extract salient features from raw data [43, 8]. While building structure into neural networks has led to state-of-the-art results across a myriad of tasks, there are significant hindrances to this approach. Indeed, these larger and more complicated models are often challenging to train and there are few guiding principles that can be used to consistently train novel architectures. As such, neural network training frequently involves large, mostly brute force, hyperparameter searches that are a significant computational burden and obfuscate scientific approaches to deep learning. Indeed, it is often unclear whether architectural additions - such as batch normalization or skip connections - are responsible for improved network performance or whether they simply ameliorate training.

There are many ways in which training a neural network can fail. Gradients can vanish or explode which makes the network either insensitive or overly sensitive to updates during stochastic gradient descent [25]. Even if the gradients are well-behaved at initialization, curvature can cause gradients to become poorly conditioned after some time which can derail training. This has led researchers to try to consider natural gradient [3] or conjugate gradient [38] techniques. While some methods like KFAC [37] are tractable, second order methods have found limited success due to the implementation challenges and computational overhead. However, quasi-second order techniques such as Adam [30] have become ubiquitous.

The choice of initial parameters, $\theta_0$, is intimately related to the initial conditioning of the gradients and therefore plays a crucial role in the success or failure of neural network training [32]. Consequently, there is a long line of research studying initialization schemes for neural networks including early seminal work by Glorot *et al.* [20] showing that the norms of the weights and biases in a fully-connected network controls whether gradients explode or vanish on average. Subsequent work by Saxe *et al.* [51] showed that the gradient fluctuations could additionally be controlled in deep linear networks. More recent contributions have included initialization schemes for fully-connected networks [52, 46], residual networks [23, 58, 63], convolutional networks [57], recurrent networks with gating [9, 19], and batch normalization [59]. While this research has found success, the analysis is often sophisticated, requiring significant expertise, and depends crucially on the architecture and choice of activation functions. An automated approach to initialization would reduce the amount of expertise necessary to train deep networks successfully and would be applicable to novel architectures. However, no objective has been identified that works for a broader range of architectures. For example, orthogonal initialization schemes identified in [51] fail in combination with ReLU nonlinearities [46] and LSUV [41] is not compatible with pre-activation residual networks [63].

In this work, we propose a strategy to automatically identify good initial parameters of machine learning models. To do this we first propose a quantity, called the gradient quotient, that measures the change in the gradient of a function after a single step of gradient descent. We argue that low gradient quotient correlates with a number of recently identified predictors of trainability including the conditioning of the hessian [18], the Fisher Information [3], and of the neural tangent kernel [28, 17, 34]. We then introduce the MetaInit (Meta Initialize) algorithm that minimizes the gradient quotient using gradieng descent to tune the norms of the initial weight matrices. We show that for two key architecture families (vanilla CNNs and Resnets), MetaInit can automatically correct several bad initializations. Moreover, we show that by initializing using MetaInit we can initialize deep networks that reach state-of-the-art results without normalization layers (e.g. batch normalization) and near state-of-the-art without residual connections. Finally, we show that MetaInit is efficient enough that it can be applied to large-scale benchmarks such as Imagenet [12].

## 2 MetaInit: Initializing by searching for less curvy starting regions

In this section, we propose an algorithm called MetaInit that adjusts the norms of the parameters at initialization so they are favorable to learning. To do so we must first identify a principle for good initialization that can be formalized into an objective function. This objective function should have other crucial properties such as being efficient to compute and easily amenable to minimization by gradient descent. These requirements rule out well-known quantities such as the condition number of the Hessian and led to the development of a novel criterion.

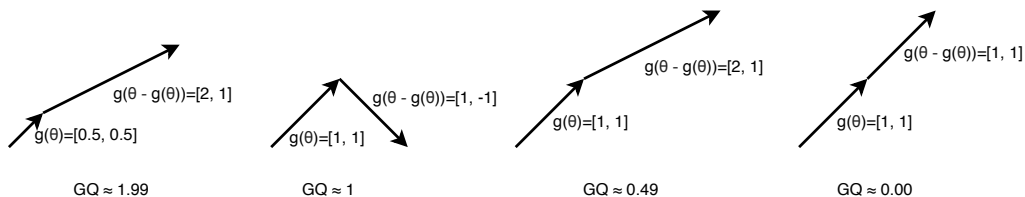

Figure 1: Illustration of the gradient quotient for different initial trajectories.

Recall that gradient descent is a first order algorithm that does not take the curvature of the function into account at each step. As discussed above, a longstanding goal in deep learning is to develop tractable optimization frameworks to try to take into account second-order information. Absent such methods, we hypothesize that a favorable inductive bias for initialization is to start learning in a region where the gradient is less affected by curvature. In this region, the magnitude and direction of the gradient should not change too abruptly due to second order effects. This hypothesis is motivated by Pennington *et al.* [46] that observed better gradient conditioning and trainability as networks become more linear, Balduzzi *et al.* [5] that proposed a successful "looks-linear" initialization for rectified linear layers, and Philipp *et al.* [47] who showed correlation between generalization and "nonlinearity".

Accordingly, consider parameters $\theta \in \mathbb{R}^N$ for a network, along with a loss function $\ell(\mathbf{x}; \theta)$. We can compute the average loss over a batch of examples, $L(\theta) = \mathbb{E}_x[\ell(x; \theta)]$, along with the gradient $\mathbf{g}(\theta) = \nabla L(\theta)$ and Hessian $\mathbf{H}(\theta) = \nabla^2 L(\theta)$. We would like to construct a quantity that measures the effect of curvature near $\theta$ without the considerable expense of computing the full Hessian. To that end, we introduce the gradient quotient,

$$\text{GQ}(L, \theta) = \frac{1}{N} \left\| \frac{\mathbf{g}(\theta) - \mathbf{H}(\theta)\mathbf{g}(\theta)}{\mathbf{g}(\theta) + \epsilon} - 1 \right\|_1 \approx \frac{1}{N} \left\| \frac{\mathbf{g}(\theta - \mathbf{g}(\theta))}{\mathbf{g}(\theta) + \epsilon} - 1 \right\|_1 \tag{1}$$

where $\epsilon = \epsilon_0(2_{\mathbf{g}(\theta) \geq 0} - 1)$ computes a damping factor with the right sign for each element, $\epsilon_0$ is a small constant and $\|\cdot\|_1$ is the L1 vector norm. As its name suggests, the gradient quotient is the relative per-parameter change in the gradient after a single step of gradient descent. We find that the step-size has virtually no effect on the gradient quotient aside from a trivial scaling factor and so we set it to 1 without a loss of generality. Parameters that cause the gradient to change explosively have a large gradient quotient, while parameters that cause vanishing gradients do not minimize this criterion since $\mathbf{g}(\theta) = \mathbf{g}(\theta - \mathbf{g}(\theta)) = 0 \implies \text{GQ}(L, \theta) = 1$ if $\epsilon > 0$. By contrast, it is clear that the optimal gradient quotient of 0 is approached when $L(\theta)$ is nearly a linear function so that $\mathbf{H}(\theta) \approx \mathbf{0}$.

**Relationship to favorable learning dynamics** The gradient quotient is intimately related to several quantities that have recently been shown to correlate with learning. Letting $\lambda_i$ be the eigenvalues of $\mathbf{H}(\theta)$ along with associated eigenvectors, $\mathbf{v}_i$, it follows that $\mathbf{g}(\theta) = \sum_i c_i \mathbf{v}_i$ for some choice of $c_i$. Furthermore, neglecting $\epsilon$ the objective simplifies to

$$\text{GQ}(L, \theta) = \frac{1}{N} \left\| \frac{\mathbf{H}(\theta)\mathbf{g}(\theta)}{\mathbf{g}(\theta)} \right\|_1 = \frac{1}{N} \sum_j \left| \frac{\sum_i \lambda_i c_i (\mathbf{e}_j^T \mathbf{v}_i)}{\sum_i c_i (\mathbf{e}_j^T \mathbf{v}_i)} \right|. \tag{2}$$

where the $\mathbf{e}_i$ are standard basis vectors. This reveals the gradient quotient is intimately related to the spectrum of the Hessian. Moreover, the gradient quotient can be minimized by either:

1. Improving the conditioning of the Hessian by concentrating its eigenvalues, $\lambda_i$, near 0.

2. Encouraging the gradient to point in the flat directions of $H$, that is to say $c_i$ should be large when $\lambda_i$ is close to zero.

There is significant evidence that improving conditioning in the above sense can lead to large improvements in learning. Notice that the Hessian, the Fisher Information, and the Neural Tangent Kernel all share approximately the same nonzero eigenvalues. From the perspective of the Hessian, the relationship between conditioning and first-order optimization is a classic topic of study in optimization theory. Most recently, it has been observed [51, 46, 18] that Hessian conditioning is intimately related to favorable learning dynamics in deep networks. In addition, it has been shown that a signature of failure in deep learning models is when the gradient concentrates on the large eigenvalues of the Hessian [21, 18]. This is precisely what condition 2 avoids. Likewise, experiments on natural gradient methods [3, 37] have shown that by taking into account the conditioning of the Fisher, one can significantly improve training dynamics. Finally, the neural tangent kernel has been shown to determine early learning dynamics [17, 34] and its conditioning is a strong predictor of trainability. In the appendix we present numerical experiments showing that this qualitative picture accurately describes improvements to a WideResnet during optimization of the gradient quotient.

**Efficiency** Computing the gradient quotient is on the same order of complexity as computing the gradient.

In addition to using the gradient quotient to measure the quality of an initial choice of parameters, it will be used as a meta-objective to learn a good initialization from a poor one as follows

$$\text{MetaInit}(L, \theta) = \arg\min_\theta \text{GQ}(L, \theta). \tag{3}$$

**Robustness as search objective** Using gradient descent on a meta-learning objective to recover from bad initialization typically would make things more difficult [36]. If the gradient vanishes for gradient descent on $L$, then it likely vanishes for a meta-learning objective that involves multiple steps on $L$ like MAML [15]. The gradient quotient avoids this problem because it is sensitive to parameters even in the presence of gradient vanishing by depending on the per-parameter values of the gradient explicitly. As such there is a "short path" between each parameter and the objective

under backpropagation. In other words, pre-training using the gradient quotient as an objective makes training more robust.

**Task agnosticity** We solve the over-fitting problems associated with meta-learning by following previous initialization approaches [52, 57] in using input data drawn completely randomly, such as $\mathbf{x} \sim \mathcal{N}(0, 1)$, during meta-initialization. We find the gradient quotient is still informative with random data and as a result the meta-initialization is largely task independent. This is not possible for other meta-learning methods which typically require large amounts of true data to compute the end-to-end training objective. It may be surprising that the gradient quotient is a good indicator even with random data, but this is consistent with previous initialization work that used random data in their analysis [20, 51] or found that good-initialization was relatively dataset agnostic [52]. We dub the use of such objectives as task agnostic meta-learning (TAML). We note that the task-agnostic nature of MetaInit implies that once a model has been initialized properly it can be used for a number of tasks.

```python
import torch

def gradient_quotient(loss, params, eps=1e-5):
    grad = torch.autograd.grad(loss,
        params, retain_graph=True, create_graph=True)
    prod = torch.autograd.grad(sum([(g**2).sum() / 2 for g in grad]),
        params, retain_graph=True, create_graph=True)

    out = sum([((g - p) / (g + eps * (2*(g >= 0).float() - 1).detach())
        - 1).abs().sum() for g, p in zip(grad, prod)])
    return out / sum([p.data.nelement() for p in params])

def metainit(model, criterion, x_size, y_size, lr=0.1,
             momentum=0.9, steps=500, eps=1e-5):
    model.eval()
    params = [p for p in model.parameters()
        if p.requires_grad and len(p.size()) >= 2]
    memory = [0] * len(params)
    for i in range(steps):
        input = torch.Tensor(*x_size).normal_(0, 1).cuda()
        target = torch.randint(0, y_size, (x_size[0],)).cuda()
        loss = criterion(model(input), target)
        gq = gradient_quotient(loss, list(model.parameters()), eps)

        grad = torch.autograd.grad(gq, params)
        for j, (p, g_all) in enumerate(zip(params, grad)):
            norm = p.data.norm().item()
            g = torch.sign((p.data * g_all).sum() / norm)
            memory[j] = momentum * memory[j] - lr * g.item()
            new_norm = norm + memory[j]
            p.data.mul_(new_norm / norm)
        print("%d/GQ = %.2f" % (i, gq.item()))
```

Figure 2: Basic Pytorch code for the MetaInit algorithm.

## 3 Implementation

The proposed meta-algorithm minimizes Equation 1 using gradient descent. This requires computing the gradient of an expression that involves gradients and a hessian-gradient product. However, gradients of the GQ can easily be obtained automatically using a framework that supports higher order automatic differentiation such as PyTorch [45], TensorFlow [1], or JAX [16]. Automatic differentiation can compute the Hessian vector product without explicitly computing the Hessian by using the identity $\nabla^2 \ell \mathbf{v} = \nabla(\nabla \ell \cdot \mathbf{v})$. The `gradient_quotient` function in Algorithm 2 provides the PyTorch code to compute Equation 1.

Like previous initialization methods [20], we find experimentally that it suffices to tune only the scale of the initial weight matrices when a reasonable random weight distribution is used - such as Gaussian or Orthogonal matrices. We can obtain the gradient with respect to the norm of a parameter $\mathbf{w}$ using the identity $\frac{\mathbf{w}}{\|\mathbf{w}\|} \cdot \nabla_{\mathbf{w}}\ell$ as in [50]. The biases are initialized to zero and are not tuned by MetaInit. As discussed above, the gradient quotient objective is designed to ameliorate issues with vanishing and exploding gradients. Nonetheless, for the most pathological initialization schemes, more help is needed. To that end we optimize Equation 1 using signSGD [7], which performs gradient descent with the sign of the gradient elements. We find that using only the sign of the gradient prevents a single large gradient from derailing optimization and also guarantees that vanishing gradients still result in non-negligible steps. The `metainit` function in Algorithm 2 provides the PyTorch code to perform Equation 3.

We find that successfully running the meta-algorithm for a Resnet-50 model on Imagenet takes 11 minutes on 8 Nvidia V100 GPUs. This represents a little less than 1% of the training time for that model using our training setup. Though we expect that recovering from initializations worse than those we consider in Section 4 could require more meta-training time.

# 4    Experiments

In this section, we examine the behavior of MetaInit algorithm across different architectures. We consider plain CNNs, WideResnets [60], and Resnet-50 [24]. Here plain networks refer to networks without skip connections that are otherwise the same as WideResnet. In order to isolate the effect of initialization, unless explicitly noted, we consider networks without normalization (e.g. batch normalization) - which has been shown to make networks less sensitive to the initial parameters. To remedy the fact that, without normalization, layers have sightly fewer parameters, we introduce a scalar multiplier initialized at 1 every two layers as in [63]. The networks without normalization do not have biases in the convolutional layers. Unless otherwise noted, we use Algorithm 2 with the default hyper-parameters.

## 4.1    Minimizing the gradient quotient corrects bad initial parameter norms

In this section, we evaluate the ability of metainit to correct bad initializations. For each architecture, we evaluate two bad initializations: one where the magnitude of the initial parameters are too small and one where they are too big. We then tune the norms of the initial parameters with the meta-algorithm. We perform experiments with 28-layer deep linear networks so as to remove the confounding factor of the activation function. The loss surface is still non-linear due to the cross-entropy loss. We use the default meta-hyper-parameters except for the number of steps, which is set to 1000, and the momentum, which is set to 0.5. As discussed above, we use randomly generated data composed of $128 \times 3 \times 32 \times 32$ input matrices and 10-dimensional multinomial targets. We evaluate the method by comparing the norms of the weights at initialization and after meta-optimization with a reference initialization that is known to perform well for that architecture.

We plot the norm of the weight matrices before-and-after MetaInit as a function of layer for each initialization protocol outlined above in Figure 3. In these experiments Gaussian(0, $\sigma^2$) refers to sampling the weight matrices from a Gaussian with fixed standard deviation $\sigma$, Fixup (Nonzero) refers to a Fixup initialization [63] where none of the parameters are initialized to zero. Gaussian(0, $\sigma^2$) is a bad initialization that has nonetheless been used in influential papers like [31]. We see that MetaInit adjusts the norms of the initial parameters close to a known, good, initialization for both the plain and residual architectures considered.

This is surprising because MetaInit does not specifically try to replicate any particular known initialization and simply ensures that we start in a region with small curvature parallel to the gradient. Though automatic initialization learns to match known initializations for certain models, we observe that it tends to differ when non-linear activation functions are used. This is expected since the aim is for the method to find new initializations when existing approaches aren't appropriate. For these types of network, we will evaluate the method through training in the next section.

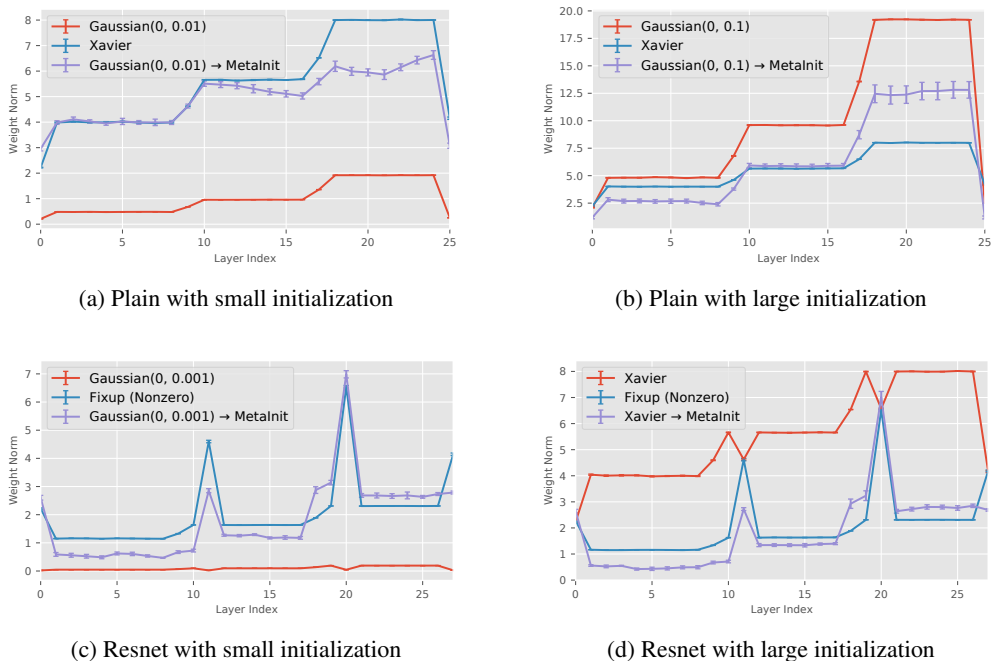

(a) Plain with small initialization

(b) Plain with large initialization

(c) Resnet with small initialization

(d) Resnet with large initialization

Figure 3: Norm of the weight matrices of a 28 layer linear network for a bad initialization (red), MetaInit applied to correct the bad initialization (purple) and a reference good initialization (blue). Note that the norm increases with the index because the number of channels in the weights increases. These results report the average and standard error over 10 trials for each random initialization. We observe that MetaInit learns norms close to known good initializations even when starting from a bad initialization.

| Model | Method | Gradient Quotient | Test Error (%) |
|---|---|---|---|
| Plain 28-10 | Batchnorm | - | 6.0 |
| | LSUV | - | 3.7 |
| | DeltaOrthogonal (Untuned) | 1.00 | 90.0 |
| | DeltaOrthogonal → MetaInit | 0.54 | 3.7 |
| WideResnet 202-4 | Batchnorm | - | 3.4 |
| | LSUV | - | 6.9 |
| | DeltaOrthogonal (Untuned) | 2.72 | 6.7 |
| | DeltaOrthogonal → MetaInit | 0.53 | 3.8 |
| WideResnet 28-10 | Batchnorm | - | 2.8 |
| | LSUV | - | 4.8 |
| | DeltaOrthogonal (Untuned) | 0.87 | 3.2 |
| | DeltaOrthogonal → MetaInit | 0.45 | 2.9 |

Table 1: Test accuracy on CIFAR-10 with the different methods. The gradient quotient reported here is computed before training. Using MetaInit to improve the initialization allows training networks that are competitive to networks with Batchnorm.

## 4.2 Improving bad initialization with MetaInit helps training

In this section, we evaluate networks trained from our meta-initialization on challenging benchmark problems. Many works in deep learning, such as [24, 10, 49, 53], have tended to treat initializations like Kaiming [23] and Orthogonal [51] as standards that can be used with little tuning to the architecture. To illustrate the need to tune the initialization based on architecture, we will compare with an untuned DeltaOrthogonal initialization [57], which is a state-of-the-art extension of the Orthogonal initialization [51] to convolutional networks. Since we hope to show that MetaInit can

automatically find good initializations, we do not multiply the initial parameter values by a scaling factor that is derived using expert knowledge based on the architecture.

It should be noted that the tuning done by MetaInit could also be derived manually but this would require careful expert work. These experiments demonstrate that automating this process as described here can be effective and comparatively easier.

**CIFAR** We use the $\beta_t$-Swish$(x) = \sigma(\beta_t x)x$ activation function [49], with $\beta_0$ initialized to $0$ to make the activation linear at initialization [5]. We use Mixup [62] with $\alpha = 1$ to regularize all models, combined with Dropout with rate $0.2$ for residual networks. For plain networks without normalization, we use gradient norm clipping with the maximum norm set to $1$ [11]. We use a cosine learning rate schedule [35] with a single cycle and follow the setup described in that paper. We chose this learning rate schedule because it reliably produces state-of-the-art results, while removing hyper-parameters compared to the stepwise schedule. All methods use an initial learning rate of $0.1$, except LSUV which required lower learning rates of $0.01$ and $0.001$ for WideResnet 28-10 and WideResnet 202-4 respectively. LSUV also uses DeltaOrthogonal initialization in convolutional layers for fairness since it is an improvement over Orthogonal initialization. The batch size used for the meta-algorithm is $32$. The number of meta-algorithm steps for WideResnet 202-4 was reduced to $200$ for this specific model. Apart from this, we use the default meta-hyper-parameters.

Table 1 shows the results of training with the various methods on CIFAR-10. We observe that without tuning DeltaOrthogonal initialization does not generalize well to different architectures. The key issue in the plain architecture case is that the effect of the activation function was not taken into account. In our training setup, the $\beta_0$-Swish $= \sigma(0)\mathbf{x} = 0.5\mathbf{x}$ results in a multiplicative scaling factor of $^1/_2$ at every layer due to the initialization of $\beta_0$ at $0$. This results in vanishing gradient in the plain architecture case. Surprisingly, while this gain is bad for plain networks it helps training for residual networks. As explained by [58, 63], downscaling is necessary for residual networks to prevent explosion. However, typically the scaling factor should be inversely proportional to the depth of the network. By contrast, the naive initialization here uses a constant gain factor. Accordingly, we observe that DeltaOrthogonal with this setup does not work well when we increase the number of layers in the network - with the accuracy decreasing by 3%. The failure of the untuned DeltaOrthogonal initialization in this setup demonstrates that the initialization must change with the architecture of the network. By contrast, using MetaInit we are able to recover strong results independent of architecture.

Our results also show that adapting the DeltaOrthogonal initialization using MetaInit leads to accuracies that exceed or are competitive to applying Batchnorm to the architectures considered here. The gap between the meta-algorithm and Batchnorm for plain networks further corroborate the theoretical results of [59], which showed that Batchnorm is not well-suited for networks without residual connections. This suggests that in general Batchnorm should not be relied upon as much as it has to correct mistakes in initialization. As a case in point, our results demonstrate that plain deep networks can be much more competitive with Resnets than is commonly assumed when a proper initialization and training setup is used. By comparison, the network with Batchnorm reaches an accuracy that is at least 2% lower for this setup. Aside from proper initialization, the key components to achieving this result for plain networks are the use of the $\beta$-Swish activation and clipping. As a reference, when we combine BatchNorm and MetaInit for the WideResnet 28-10, we obtain the same performance as BatchNorm by itself (2.8%). This is not unexpected since BatchNorm makes the network more robust to initialization.

LSUV [41] is a data-dependent initialization method that tries to mimic BatchNorm by normalizing the layers at initialization. Our results show that this approach improves results for plain networks, but LSUV actually make results worse than the naive initialization for WideResnet. This failure is consistent with the fact that LSUV cannot scale the residual layers by depth, which was shown to be crucial for stability as in [58, 63]. As a result, LSUV requires using lower learning rates than the other methods discussed here to prevent divergence.

**Imagenet** We use the Resnet-50 architecture with scalar bias before and after each convolution following [63]. In order to showcase the importance of adapting the initialisation to the architecture we will consider two activation functions - Swish $= \sigma(\mathbf{x})\mathbf{x}$ and ReLU. We use the same training setup and hyper-parameters as [63] - except for the initialization which is set to DeltaOrthogonal. The application of MetaInit was much less straightforward than in the previous case due to the complexity of the model considered. In order to obtain a good estimate of the gradient quotient, we had to use a batch size of $4096$ examples. This required using smaller random inputs, of size $32 \times 32$,

| Activation | BatchNorm* | GroupNorm* | Fixup* | DeltaOrthogonal (Untuned) | MetaInit |
|---|---|---|---|---|---|
| ReLU | 23.3 | 23.9 | 24.0 | 24.3 | 24.6 |
| Swish | - | - | - | 99.9 | 24.0 |

Table 2: Top-1 Test error on Imagenet ILSVRC2012 for a Resnet-50 model with different methods. * The columns for BatchNorm, GroupNorm and Fixup are reference results taken directly from [63]. They are not initialized in the same way and so are not as directly comparable. MetaInit produces more consistent results than untuned DeltaOrthogonal initialization as we vary the activation functions.

compared with the size of Imagenet images while meta-initializing. Furthermore, it was necessary to use cross-validation to select the momentum parameter for the meta-algorithm between the values of $0.5$ and $0.9$.

Table 2 shows that using MetaInit leads to more consistent results as we change the architecture compared to untuned DeltaOrthogonal initialization. In this case, the change in architecture is driven by the choice of activation function. As first noted by [23], ReLU activation layers downscale the standard deviation of pre-activations by $1/\sqrt{2}$; by contrast the Swish activation leads to a reduction in the standard deviation of about $1/2$ at each layer. Coincidentally, the downscaling provided by the ReLU works well for this specific architecture, while that one implied by the Swish is too large and prevents learning. However, with MetaInit, we observe training in both cases. Moreover, our results are competitive with BatchNorm and GroupNorm. We believe that the gap in performance in BatchNorm is mainly due to the regularization properties of BatchNorm. We view it is a success of our method that we are able to disentangle trainability and generalization and quantify the regularizing role of BatchNorm. Our results are also competitive to the Fixup [63] initialization method, which was developed for residual networks with positive homogeneous activations - like ReLU units but unlike Swish.

## 4.3 Ablation

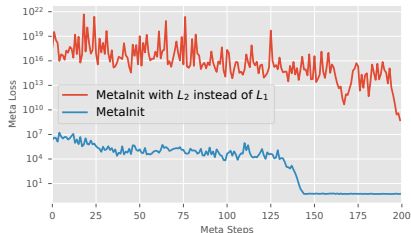 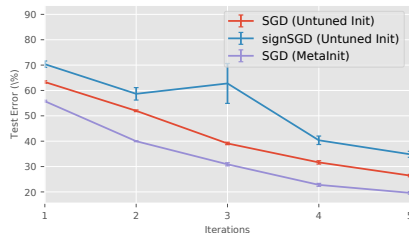

Figure 4: Meta-Learning curves for MetaInit using $L_2$ norm in Equation 1 instead $L_1$. The results reported are averaged over 10 trials. Using the $L_1$ norm leads to faster minimization.

Figure 5: Learning curve of WideResnet 16-4 on CIFAR-10 comparing SGD with MetaInit initialization to SGD and signSGD with untuned initialization during supervised training. MetaInit leads to faster convergence.

We evaluate the importance of using the $L_1$ norm in Equation 1 during the meta-optimization phase. We will consider a linear plain network with 28 layers and width 1 with a bad initialization sampled from Gaussian$(0, 0.1)$. The random inputs have size $128 \times 3 \times 32 \times 32$ with 10 classes and the momentum hyper-parameter set to $0.5$. Figure 4 demonstrates the importance of the $L_1$.

Next we evaluate how the proposed meta-initialization compares to using signSGD directly during regular training to mitigate bad initializations. Figure 5 shows results on CIFAR-10 training for 5 supervised epochs with cosine learning rate. SGD and MetaInit were both trained using a learning rate of $0.1$ while the learning rate for signSGD had to be reduced to $0.001$ to avoid divergence. Unlike signSGD, MetaInit discovers a good initialization without using any supervised data. signSGD must recover from the bad initialization using supervised updates, which could and does influence convergence in this case.

# 5   Limitations

The algorithm proposed in this paper has limitations, which we hope will be addressed in future work:

1. In some cases, applying MetaInit successfully requires tuning the meta-hyper-parameters. However, the number of hyper-parameters added is small compared to directly cross-validating over different initial norms for each parameter.

2. The proposed algorithm uses gradient descent to help gradient descent. However, the gradient descent process on the meta objective can and does fail for very large scale models and input dimensions. Getting good estimates of the gradient and Hessian-vector product can require very large batch sizes for big models.

3. Improving initialization does not necessarily address all numerical instability problems. In particular, training with lower precision without normalization is particularly unstable.

4. We learn the norms of the parameters, not the full parameters. This mirrors many influential initializations, but it is possible to imagine that certain architectures might require adapting all initial parameters. Thus, for the moment, selecting the initial distribution of parameters still requires expert intervention.

# 6   Additional Related Work

There are a number of research directions that are related to MetaInit aside from the work on optimizers, normalization methods, and initialization schemes already discussed above. Maclaurin *et al.* [36] performed gradient descent on hyper-parameters by propagating gradients through the entire training procedure. While this approach can be used to tune the parameter norms, it has a significant computational cost, can lead to overfitting, and gradients taken through optimization can be very poorly conditioned [40]. MAML [15] is a related meta-algorithm that searches for weight initializations that are beneficial to few-shot learning. While it also produces a weight initialization, it is not mutually exclusive with the proposed approach due to their different focus. For example, MetaInit could be used as the initialization for MAML.

As discussed briefly above there has been a long line of significant work to improve optimizers to be robust to poor initialization. Several examples of this include momentum [48, 42], RMSProp [], ADAM and ADAMAX [30], ADADELTA [61], or NADAM [13]. Optimizers that exploit curvature information such as [14] have been proposed but they can negatively affect generalization [54]. Furthermore, recovering from a bad initialization using regular supervised training steps could still negatively affect the generalization of the model. A form of meta-learning can also be used to improve optimization such as [6], which tunes the learning rate using hypergradient descent. More recently, a natural extension of this work has focused on learning the structure of the optimizer itself using meta-learning [39]. Finally, a series of papers has used population based training [29] to identify training schedules. These powerful approaches still require a reasonable starting point for learning and, as with MAML, could be paired well with MetaInit.

As described earlier, a very successful approach to improving training robustness is adding normalization. The most successful of these approaches is arguably BatchNorm [27]. However, as first explained in [59] and supported in Section 4.2, BatchNorm does not apply well to all architectures. Other normalization methods such as LayerNorm [4] or GroupNorm [56] likewise appear to ameliorate training in some circumstances but have a deleterious effect in others cases. Finally, techniques like gradient clipping [44] are extremely useful, but require a reasonable starting point for learning. As discussed above, we observe that gradient clipping works well in combination with MetaInit.

# 7   Conclusion

We have proposed a novel method to automatically tune the initial parameter norms under the hypothesis that good initializations reduce second order effects. Our results demonstrate that this approach is useful in practice and can automatically recover from a class of bad initializations accross several architectures.

## Acknowledgements

We would like to thank David Grangier, Ben Poole, and Jascha Sohl-Dickstein for help discussions.

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
