[Supplementary Material · supp.pdf]

# Appendix

We discuss several experiments that probe the effects of minimizing gradient quotientdiscussed in section [2]. Recall eq. ([2]) related the gradient quotient to two terms, $\mathbf{H}(\theta)\mathbf{g}(\theta)$ and $\mathbf{g}(\theta)$. It was hypothesized that the gradient quotient would be minimized when the Hessian was well-conditioned and the gradient was orthogonal to the large eigen-directions of the Hessian. We additionally hypothesized that this would lead the Neural Tangent Kernel to have improved conditioning as well. In fig. [6] we track several quantities for the WideResnet

Figure 6: **Statistics of a WideResnet during MetaInit** (a) shows the gradient quotient over the course of training, (b) shows the gradient norm $||\mathbf{g}||$, (c) shows the norm of the hessian-gradient product $||\mathbf{Hg}||$, (d) shows the overlap between the gradient and the hessian-gradient product studied in [21] $\cos\theta = \frac{\mathbf{g^T Hg}}{||\mathbf{g}||||\mathbf{Hg}||}$, and (e,f) show the top eigenvalues of the NTK and the gap between the largest eigenvalue and the next-largest eigenvalue.

architecture described in the main text over the course of meta-training. We observe that all of these quantities track the gradient quotient as expected. In particular: the gradient norm decreases, the hessian-gradient product becomes smaller and the two quantities become closer to orthogonal. Moreover, the NTK becomes significantly better conditioned. Not only do the eigenvalues decrease substantially but the gap between the top eigenvalue and the next-highest eigenvalue becomes smaller. This gap has been seen to control the rate of learning in the case of an MSE loss [34].