[Reviews · NeurIPS 2019]

Reviewer 1



Update: The authors have addressed my questions. I hope in the camera ready there is a clear discussion on Taylor expansion VS. finite difference (at least in the appendix). I also second the other reviewer on the importance of comparing in the batch norm case, since the method should be used as a general purpose initialization scheme. Longer Summary: - Authors introduce the GradientDeviation criterion, which characterizes how much gradient changes after gradient step. Simple and avoids full Hessian (+) - They use meta-learning to learn the scale of initialization such that GradientDeviation is minimized. - They claim meta-learning the scale is mostly architecture-dependent can be done with random data and random labels, without the need of using a specific dataset (+) - They compare with other initializations schemes (DeltaOrthogonal, LSUV) and with Batchnorm. Beats other schemes and competitive with Batchnorm (+) Shortcomings: - Can be problematic for very large models (-) - Can only learn scale of parameters (but from experience I think the actual distribution does not matter too much anyways for independent initializations) The authors claim their approach is more general than analytical approaches. Analytical approaches do not account for nonlinearity. Originality: Combines two-lines of work: learning to initialize and developing analytic initializations (e.g. Xavier, Kaiming). Quality: Enough experiments, paper well written. Related section seems a bit thin. Clarity: The paper is very well written and easy to read. Significance: Their "GradientDeviation" criterion can be reused and explored by others. Their method can be combined with usual initialization schemes.

Reviewer 2



They propose a novel algorithm that automatically finds the good initialization for a neural network by meta-learning methodology even without specific data (i.e. in a domain-independent way). To do that, they propose a new metric that is called gradient deviation, which measures the scale of one step ahead gradient likewise in MAML. They assume that the gradient of good initial parameters is less affected by the curvature near it. I think It’s nice to bring the idea of meta learning into learning the initial parameters of model and the objective function. But it’s lack of evidence that the method is working in a sense that there is no theoretical proof of experimental results. Q1. I’m not the expert of optimization theory, so I can’t fully confirm that the hypothesis you made is valid or not. Is there any theory that plain surface in the initial point leads to a better local minima? Even if you have no theoretical proof, there should be an experimental support that Gradient Deviation represents a metric of good initialization (i.e. Gradient Deviation – final performance for various initializations of the model) Q2. I don’t understand the protocols you used in experiments. Why do you have to remove skip connections and batch normalization layers? Is it natural to compare random init (or other known initialization method) vs. meta-learned init?

Reviewer 3



This is a clear well written paper on using meta learning to learn better initial parameters for training deep neural networks. It uses automatic differentiation to optimize a ratio of magnitude of gradient change in order to learn scale value for initial parameters for different layers. While it is a simple algorithm, figure 2 is very interesting. However, why not also show results for non-random data? The paper also mention it operates in 'data-agnoistic' fashion, what advantages does it bring to be data agnostic? If the algorithm is truly data agnostic, are there results on how scale learned for a network can transfer to multiple datasets? The paper makes the statement regarding 'less curvy' starting regions. However the loss in equation 2 is only looking at the magnitude of the gradient and not necessarily the curvature, due to the absolute value taken. - What happens if metainit is combined with batchnorm? - Can you show a training error plot as function of update iterations? that would be helpful to compare with metainit vs baseline optimization.

[Author Response · NeurIPS 2019]

We would like to thank the reviewers for their helpful feedback; we will use it to significantly improve our manuscript.

**Reviewer 2**

We would like to give a high level summary of our paper for clarity. Neural networks are hard to train and techniques,
like skip connections and batch normalization, have been developed to make training easier, but only work in some
cases and add complexity to the architecture. Simultaneously, there has been recent (and not so recent) theoretical work
showing that properly scaling the weights in a network at initialization can significantly improve learning. Unfortunately
the "correct" weight scale is highly architecture dependent. The main goal of this work is to automate this process.

1. *"Q2. I don't understand the protocols you used in experiments. [...] Is it natural to compare random init (or*
*other known initialization method) vs. meta-learned init?"*: Since we are trying to automate the process of
choosing a good initialization, it is natural and important to compare MetaInit to handcrafted initializations.
In addition, we also compare with, to our knowledge, the best automated initialization method called LSUV
(Mishkin et al, 2015) in Table 1. We outperform LSUV by at least 2% on CIFAR-10 for residual networks.

2. *"Why do you have to remove skip connections and batch normalization layers?"*: An important question is
whether or not the aforementioned architectural tricks are necessary for training models that perform well
or can we get by with good initialization alone? Previously, this had been difficult to study since, as shown
in Table 1, performance on vanilla architectures using traditional initialization schemes does not succeed.
However, using MetaInit we are able to substantially close the gap and compare architectural features directly.
We believe being able to do this efficiently is necessary to make machine learning more rigorous.

3. *"Q1. [...] Even if you have no theoretical proof, there should be an experimental support"'*: We would like to
point out that there is a large body of work relating initialization to trainability dating back at least to (Glorot
and Bengio, 2010) and an even longer history relating Hessian conditioning to learning (Nocedal et al, 1998).
We do provide strong experimental support that the gradient deviation is a good metric. Table 1 and 2 show
that minimizing the gradient deviation can improve the performance up to 60% for some models, even on
Imagenet. However, we agree with the referee that can be even more explicit in showing that gradient deviation
is strongly correlated with test-time performance. To that end we will add gradient deviation measurements
along with test error to all of our tables. For example,

| Model | Method | Test Error (%) | Gradient Deviation |
|---|---|---|---|
| WideResnet 204-4 | DeltaOrthogonal | 6.7 | 2.29 |
| | MetaInit | 3.4 | 0.50 |

**Reviewer 3**

Thanks for your feedback, we will add the requested experiments.

1. *"However, why not also show results for non-random data? The paper also mention it operates in 'data-*
*agnoistic' fashion, what advantages does it bring to be data agnostic?"*: We confirm that the algorithm also
works well with non-random data and we will add an experiment to that effect. Non-random labelled data is
more costly than random data so it is useful when a method can work without it (i.e. for few-shot learning).

2. *"equation 2 is [..] not necessarily [looking at] the curvature"*: Equation 4 shows how the gradient deviation is
related to the curvature (eigenvalues) of the Hessian. The gradient deviation is upper-bounded by a weighted
average of the eigenvalues, with higher weights in the dimension where the gradient is higher. Note, that the
gradient deviation does not go to zero if the magnitude of the gradient vanishes.

3. "What happens if metainit is combined with batchnorm?": We find that MetaInit still works with batchnorm,
we will add these experiments.

4. *"Can you show a training error plot as function of update iterations?"*: We will add these curves, which show
it usually converges much faster than the baseline.

**Reviewer 1**

1. *"Derive the analytical scaling factors [...] to compare with classic initializations (Xavier, Kaiming)"*: Figure
2 compares the scaling factors found by MetaInit with those found by Xavier and Fixup initialization. These
experiments show that the MetaInit scaling factors match those of Xavier for architectures where Xavier
initialization is appropriate. We will include a figure comparing with Kaiming in the appendix.

2. *"The related work section seems a bit thin"*: We plan to use part of the additional space allotted to expand the
related work to include MAML, hyper-gradients and traditional initializations.

3. *"Compute the criterion using finite-difference instead of Taylor expansion and see which is better."*: Thank
you for the suggestion, we have started comparing the two approaches and will include a discussion.

[Meta-Review · NeurIPS 2019]

After the author rebuttal period the reviewers converged upon a clearer acceptance recommendation. The authors were able to point to ways in which the experiments in the paper supported their line of analysis. The original paper was also perceived as being generally well written and the approach and experiments are likely to be of interest to the community.